# FedANC: Adaptive Sparse Noise Scheduling for Federated Differential Privacy

## Abstract

Federated Learning (FL) enables multiple clients to collaboratively train a shared model without sharing raw data. Although this reduces direct exposure of local data, model updates can still leak sensitive information through gradient-based attacks. Differential Privacy (DP) mitigates this risk by adding calibrated noise to updates, providing formal guarantees. However, most existing DP-FL methods adopt fixed noise scales and uniform injection across all gradient dimensions, without adapting to client heterogeneity or training dynamics. This often results in poor privacy-utility trade-offs. To overcome these limitations, we propose **FedANC**, an adaptive differential privacy framework for FL. It consists of three components: (i) an *Adaptive Noise Controller* (ANC) with an LSTM-based design that generates client-specific noise scales and sparsity ratios from local training feedback; (ii) a *Selective Noise Injection* mechanism that perturbs only the most sensitive gradient entries; and (iii) a *Privacy Budget Regularization* term that aligns per-round updates with a predefined privacy target. For stability, the ANC is pretrained with synthetic feedback that simulates typical training behavior. We provide theoretical guarantees on both convergence and differential privacy. Extensive experiments demonstrate that FedANC achieves higher accuracy, faster convergence, and stronger privacy protection compared with existing approaches.

## 1 Introduction

Federated Learning (FL) (McMahan et al., 2017; Nguyen et al., 2021) enables multiple clients to collaboratively train a shared model without exchanging their raw data. This decentralized paradigm is particularly valuable in privacy-sensitive domains such as voice recognition, healthcare, and human activity monitoring (Yu et al., 2020; Khan et al., 2021; Cui et al., 2021). Although FL reduces data exposure by keeping training data local, it does not guarantee protection against information leakage. Recent studies have demonstrated that adversaries can reconstruct sensitive user data by analyzing shared gradients. Techniques such as DLG (Zhu et al., 2019), gradient inversion (Geiping et al., 2020), and batch-level leakage analysis (Yin et al., 2021) exploit high-magnitude gradient components, which often encode fine-grained and data-specific patterns.

Differential Privacy (DP) provides a principled way to mitigate such leakage risks by adding calibrated noise to model updates (Dwork & Roth, 2014). In deep learning, DP-SGD (Abadi et al., 2016) adds noise after clipping gradients to a fixed bound, and DP-FedAvg (McMahan et al., 2018) extends this approach to FL, enabling user-level privacy guarantees. However, most DP-FL methods rely on static configurations, where the same clipping bound and noise scale are applied to all clients and rounds. This uniform setting does not reflect the heterogeneous data distributions, convergence patterns, and privacy needs that are common in real federated deployments.

Another limitation comes from the way noise is injected. Many existing methods perturb all gradient coordinates equally using isotropic Gaussian noise (Zhu et al., 2019; Zhao et al., 2020). This approach ignores the varying informativeness of different components. As a result, low-magnitude gradients, which usually carry less sensitive information, are distorted as heavily as high-magnitude ones. This unnecessary perturbation not only degrades utility but also worsens the privacy-utility trade-off, especially for complex models and resource-constrained clients.

A further challenge is the lack of mechanisms to dynamically control how privacy loss evolves during training. While some recent studies propose adaptive or non-uniform budget allocation strategies (Li

et al., 2022; Kiani et al., 2025), they often rely on heuristics or centralized scheduling. Such designs limit scalability and fail to adapt to client-specific dynamics in real federated environments.

To address these limitations, we propose **FEDANC** (**Fed**erated **A**daptive **N**oise **C**ontrol), a unified framework designed to support efficient and personalized differential privacy in federated learning. As illustrated in Figure 1, FEDANC consists of three tightly coupled components. These components jointly adapt the privacy mechanism to client-specific training dynamics while ensuring compliance with a global privacy budget.

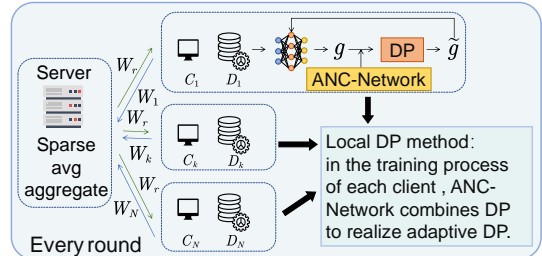

Figure 1: Design of FEDANC Framework.

The first component, the *Adaptive Noise Controller* (ANC), is a client-side module based on a lightweight network. It generates two key privacy parameters: the noise scale $\beta_t$ and the sparsity ratio $\gamma_t$. These values are computed using local gradient norms and training loss, enabling temporally adaptive and personalized noise control (Wu et al., 2022). The second component, the *Selective Noise Injection* module, perturbs only a subset of gradient entries identified as most informative. The sparsity ratio $\gamma_t$ determines the number of perturbed coordinates, allowing the system to focus noise on sensitive components while preserving useful information in the remaining gradients (Dai et al., 2022). To enforce formal privacy guarantees, the third component introduces a *Privacy Budget Regularization* term into the training objective. This term penalizes deviations from a predefined per-round budget and provides differentiable feedback to the ANC, enabling automatic adjustment of privacy parameters without manual tuning. To improve stability during early training, the ANC is pretrained using synthetic input patterns that emulate common gradient behaviors observed.

Building on the above insights, this work makes the following key contributions: (1) We introduce FEDANC, a unified and learnable framework that supports dynamic, client-specific differential privacy; (2) We present ANC, a lightweight controller that enables real-time privacy adaptation based on local training dynamics; (3) We provide a selective gradient perturbation strategy that improves privacy-utility trade-offs by focusing noise on high-sensitivity components; (4) We design a differentiable regularization objective that enforces privacy budget constraints during training.

## 2 BACKGROUND AND MOTIVATION

Federated learning is a distributed paradigm where a set of clients $\{1, \ldots, N\}$ collaboratively train a global model without sharing their local data. Formally, the global training objective is:

$$\min_{\mathbf{w}} \mathcal{L}(\mathbf{w}, \mathcal{D}) = \sum_{i=1}^{N} \frac{|\mathcal{D}_i|}{|\mathcal{D}|} \mathcal{L}_i(\mathbf{w}, \mathcal{D}_i), \tag{1}$$

where $\mathbf{w}$ denotes the global model parameters and $\mathcal{D} = \cup_{i=1}^{N} \mathcal{D}_i$ is the union of all local datasets. A widely used algorithm to solve this objective is FedAvg (McMahan et al., 2017), which proceeds in communication rounds by sampling clients, performing local training, and aggregating updates from selected clients on the server. Although FL avoids direct data sharing, model updates can still leak sensitive information. Differential privacy (Dwork & Roth, 2014) is often employed to mitigate this risk. A randomized mechanism $\mathcal{M}$ satisfies $(\varepsilon, \delta)$-DP if, for any neighboring datasets $D$ and $D'$, and any measurable set $S$, it holds that:

$$\Pr[\mathcal{M}(D) \in S] \leq e^{\varepsilon} \Pr[\mathcal{M}(D') \in S] + \delta, \tag{2}$$

where $\varepsilon$ quantifies the privacy guarantee and $\delta$ denotes the failure probability. A standard tool is the Gaussian mechanism, which releases $f(D) + \mathcal{N}(0, \sigma^2 \mathbf{I}_d)$ with noise scale $\sigma$ determined by the $\ell_2$-sensitivity $\Delta_2 f$:

$$\sigma \geq \frac{\Delta_2 f}{\varepsilon} \sqrt{2 \ln \left( \frac{1.25}{\delta} \right)}. \tag{3}$$

In FL, local differential privacy (LDP) is typically enforced by perturbing client updates before transmission. Early approaches such as DP-SGD (Abadi et al., 2016) and DP-FedAvg (McMahan

et al., 2018) achieve this by clipping gradients and adding fixed Gaussian noise. While theoretically sound, these methods often degrade model accuracy because they rely on static privacy parameters that cannot adapt to client heterogeneity or evolving training dynamics.

To improve adaptability, recent works propose dynamic strategies. For instance, DPSFL (Zhang et al., 2024) adjusts clipping thresholds, (Kiani et al., 2025) adopt time-adaptive budget allocation, and (Ranaweera et al., 2025) design a wavelet-based perturbation guided by gradient structure. These studies demonstrate that adaptive mechanisms can better balance privacy and utility. At the same time, gradient leakage remains a serious challenge. Gradient inversion attacks (Zhu et al., 2019; Geiping et al., 2020) reveal that high-magnitude gradient entries can expose sensitive data features. To address this, prior work (Fu et al., 2021; Zhang et al., 2022) explores clipping and targeted perturbation, showing that focusing noise on informative components improves robustness.

In addition to DP, secure aggregation protocols (Bonawitz et al., 2017) prevent the server from accessing individual updates during communication. Meanwhile, DP-FL has been applied in domains such as clinical prediction (Shukla et al., 2025), speech recognition (Pelikan et al., 2023), and FL simulation (Granqvist et al., 2024). Recent surveys (Kairouz et al., 2021; Shan et al., 2024) emphasize the importance of accurate privacy accounting, calibrated clipping, and adaptive noise strategies, and highlight the problem of designing personalized mechanisms that reflect each client's conditions.

Despite these advances, most existing methods still employ fixed or heuristic schedules that remain unchanged across clients. They also inject noise uniformly across all gradient coordinates, which limits flexibility and leads to suboptimal trade-offs. Motivated by these gaps, we propose FEDANC.

## 3 DESIGN OF PROPOSED FEDANC

To address these limitations identified above, we propose the FEDANC framework, which combines three main components: (i) an LSTM-based adaptive noise controller that adjusts privacy parameters from local feedback, (ii) a sparse gradient perturbation module that injects noise into the most sensitive coordinates, and (iii) a privacy-loss regularization term that enforces system-level privacy constraints. The following sections detail each component and their interaction in enabling efficient and privacy-preserving federated learning.

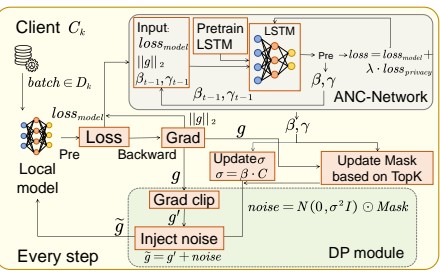

Figure 2: The proposed design of ANC.

### 3.1 DESIGN OF ADAPTIVE NOISE CONTROLLER

As shown in Figure 2, the controller is a core component that enables adaptive privacy control. Its primary function is to generate the noise scale $\beta_t$ and the sparsity ratio $\gamma_t$ for each local training round based on training feedback.

**Controller structure.** ANC takes as input a vector of local training signals:

$$x_t = [\|g_t\|_2, \ell_t, \beta_{t-1}, \gamma_{t-1}]^\top, \tag{4}$$

where $\|g_t\|_2$ represents the norm of gradient, $\ell_t$ denotes the current training loss, and $(\beta_{t-1}, \gamma_{t-1})$ are the privacy parameters from the previous round. These inputs reflect both current training dynamics and historical information. The vector $x_t$ is processed by an LSTM (Kim et al., 2016) to capture temporal dependencies across rounds. Its hidden state $h_t$ is passed through a dropout layer and then a fully connected layer to produce intermediate outputs $(u_\beta, u_\gamma)$. The final privacy parameters are computed as:

$$\beta_t = \ln(1 + \exp(u_\beta)), \quad \gamma_t = \gamma_{\min} + (\gamma_{\max} - \gamma_{\min}) \cdot \sigma(u_\gamma), \tag{5}$$

where $\sigma(\cdot)$ is the sigmoid function, and $\gamma_{\min} = 0.1$, $\gamma_{\max} = 0.9$ define the sparsity range (Dai et al., 2022; Li et al., 2024).

**Pretraining strategy.** During federated learning, raw outputs from ANC may produce excessively large or small values for the noise scale or sparsity ratio, which can hinder convergence and degrade model performance. This problem is most critical in the early training rounds, which strongly influence final model quality (Achille et al., 2019; Yan et al., 2022; 2023).

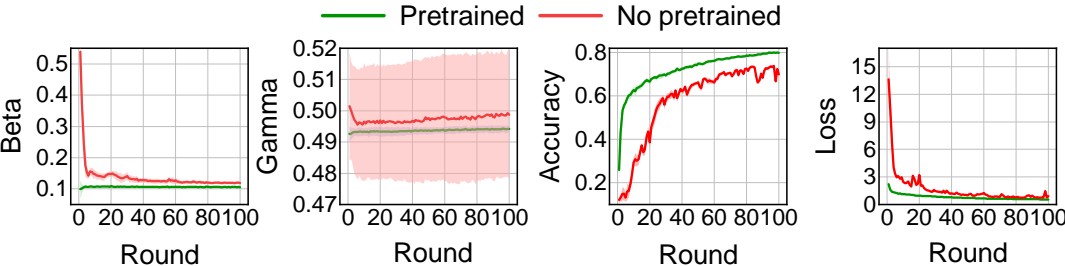

Figure 3: Comparison of ANC output stability with and without pretraining across training rounds.

To mitigate this, we pretrain ANC on synthetic data so that it generates stable and reliable privacy parameters from the beginning of training. To achieve this, we construct synthetic inputs based on typical gradient norms and loss values. Specifically, the training signals are drawn from:

$$\|g\|_2 \sim \mathcal{U}[10^{-5}, 0.1], \quad \ell \sim \mathcal{U}[0.1, 10], \tag{6}$$

These ranges reflect values commonly observed during training process, as discussed in (Abadi et al., 2016). The target privacy parameters are sampled from:

$$\beta^* \sim \mathcal{U}[0.3, 0.7], \quad \gamma^* \sim \mathcal{U}[\gamma_{\min}, \gamma_{\max}], \tag{7}$$

where $\beta^*$ defines the desired noise scale and $\gamma^*$ specifies the sparsity ratio. The range for $\beta^*$ is chosen based on empirical evidence from prior work (Talaei & Izadi, 2024; Wang et al., 2024; Kiani et al., 2025), which identifies this interval as providing effective privacy-utility trade-offs in practical federated learning scenarios.

The pretraining objective uses mean squared error loss:

$$\mathcal{L}_{\text{pre}} = \frac{1}{N} \sum_{i=1}^{N} \left[ (\beta_i - \beta_i^*)^2 + (\gamma_i - \gamma_i^*)^2 \right], \tag{8}$$

where $(\beta_i, \gamma_i)$ are the predictions generated by ANC, and $(\beta_i^*, \gamma_i^*)$ are the corresponding ground truth targets. This supervised regression process guides the controller toward stable parameter regions, which enhances convergence and robustness in downstream federated training.

Although pretraining privacy controllers remains underexplored, related work in neural architecture search (Zoph & Le, 2017) and controller-based meta-learning (Jiang et al., 2019) suggests that using synthetic supervision or proxy tasks can improve the controller's effectiveness and stability. Furthermore, adaptive differential privacy mechanisms such as AdaDPS (Li et al., 2022) also utilize auxiliary signals to adjust privacy parameters, which aligns with our design approach. These findings provide strong motivation for pretraining ANC and support its role in enabling effective and stable privacy adaptation during federated learning.

**Personalized privacy control.** After pretraining, each client runs its own instance of ANC. This allows client-specific privacy adaptation that reflects local training dynamics and data characteristics. Such personalization is particularly important in federated learning, where data distributions and model behavior vary significantly across clients.

As shown in Figure 3, our experiments on FashionMNIST, as described in *Experimental Setup* section, demonstrate that pretraining the ANC significantly enhances the stability of privacy parameter outputs, particularly in the early training rounds indicated by the shaded regions in Figures 3(a) and (b). This increased stability contributes to faster convergence and improved model performance compared to configurations using untrained controllers. By dynamically adjusting privacy parameters based on local training signals, ANC enables FEDANC to support adaptive and personalized differential privacy throughout the learning process.

### 3.2 DESIGN OF SPARSE GRADIENT PERTURBATION

To utilize the adaptive parameters produced by ANC, we adopt a *sparse noise injection strategy* that improves the trade-off between privacy protection and model utility. This strategy selectively adds noise to the most privacy-sensitive gradient components, rather than applying it uniformly.

The motivation for this design comes from recent studies (Zhu et al., 2019; Zhao et al., 2020; Dai et al., 2022), which show that gradient entries with large magnitudes often reveal more information about the input data and are more vulnerable to reconstruction-based attacks. These components contribute significantly to privacy leakage and are natural candidates for selective perturbation.

In conventional methods, Gaussian noise is applied uniformly to all gradient entries. Although this ensures differential privacy, it also perturbs low-magnitude components that contain little information and have minimal impact on privacy risk. As a result, the overall signal-to-noise ratio decreases, which can slow convergence and reduce model accuracy, especially when the privacy budget is tight.

To address this issue, we propose a gradient perturbation method that concentrates noise on the top-$k$ most informative components. The process begins by applying $\ell_2$ norm clipping to limit the sensitivity of the gradient vector. Then, we identify the top-$k$ entries with the largest magnitudes, where $k = \lceil \gamma_t d \rceil$ is computed using the sparsity ratio $\gamma_t$ and the total gradient dimension $d$. Gaussian noise $\xi$ is sampled with variance $(\beta_t C)^2$, where $C$ is the clipping bound, and is injected only into the selected entries. This is implemented using a binary mask $M \in \{0,1\}^d$, resulting in the perturbed gradient $\widetilde{g} = g' + M \circ \xi$. This selective perturbation process is closely aligned with the dynamic outputs of ANC and enables more efficient privacy protection.

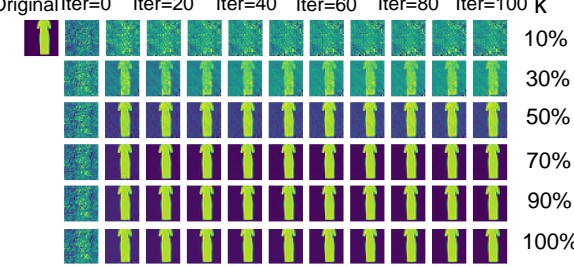

Figure 4: Top-$k$ gradients can reveal sensitive info.

By adapting the sparsity and noise scale to each client's training state, the method supports fine-grained, personalized privacy control. As shown in Figure 4, even a small subset of gradient entries (e.g., the top 50%) can be sufficient for reconstructing private data, as demonstrated in prior attack studies such as DLG (Zhu et al., 2019). These findings highlight the need for targeted protection mechanisms and validate the effectiveness of our sparse noise injection strategy.

### 3.3 Design of Privacy Budget Alignment

To complement the sparse perturbation mechanism and enforce system-wide privacy guarantees, we introduce a *privacy-loss regularization term* that guides the ANC to generate parameters consistent with a target privacy budget $\epsilon_{\text{target}}$. This component integrates naturally with the adaptive noise scale $\beta_t$ and sparsity ratio $\gamma_t$ defined previously.

The $\ell_2$ sensitivity of the top-$k$ Gaussian mechanism is given by $\Delta_2 = C\sqrt{\gamma_t d}$, where $C$ is the clipping bound and $d$ is the gradient dimension. Following the standard Gaussian mechanism (Dwork & Roth, 2014), the per-round privacy guarantee $(\epsilon_t, \delta)$ satisfies:

$$\epsilon_t = \frac{\Delta_2}{\sigma}\sqrt{2\ln(1.25/\delta)}, \tag{9}$$

where $\sigma = \beta_t C$ is the standard deviation of the injected noise. Substituting the expressions for $\Delta_2$ and $\sigma$ gives a closed-form estimate of the privacy cost:

$$\hat{\epsilon}_t = \frac{\sqrt{2\gamma_t d \ln(1.25/\delta)}}{\beta_t}. \tag{10}$$

To ensure the learned parameters $(\beta_t, \gamma_t)$ satisfy the privacy constraint $\hat{\epsilon}_t \leq \epsilon_{\text{target}}$, we define a quadratic penalty:

$$L_{\text{privacy}}^{(t)} = \left(\hat{\epsilon}_t - \epsilon_{\text{target}}\right)^2. \tag{11}$$

This term penalizes deviations from the target in a smooth and differentiable manner and is seamlessly incorporated into the total training loss:

$$\mathcal{L}_{\text{total}} = \frac{1}{|B|}\sum_{i \in B} \ell(f(\theta; x_i), y_i) + \lambda L_{\text{privacy}}^{(t)}, \tag{12}$$

where $\ell(f(\theta; x_i), y_i)$ is the prediction loss over mini-batch $B$, and $\lambda$ controls the regularization strength. In each round, the ANC receives gradient statistics and loss values, and outputs the noise scale $\beta_t$ and sparsity ratio $\gamma_t$ to control the intensity and focus of perturbation. It is trained via backpropagation using gradients from both loss terms, enabling a principled trade-off between utility and privacy. The module is lightweight(18,050 parameters, 92K FLOPs per forward pass, 5.3ms per full training step on an NVIDIA RTX 4090), model-agnostic, and integrates into standard gradient-based training.

## 3.4 Putting All Components Together

The complete workflow of the proposed FEDANC framework is provided in Algorithm 1 in Appendix A.1. FEDANC builds upon the standard FedAvg protocol and incorporates three core components that collectively support efficient, personalized, and privacy-preserving training. At the beginning of each communication round $r$, the server selects a subset of clients $S_r$ according to a ratio $p$ and and broadcasts the current global model $W_r$ to the selected clients. Each selected client sets its local model as $W_k \leftarrow W_r$ and activates its ANC module, which is initialized with the pretrained ANC if unavailable. Each client then performs $E$ local training epochs. For each mini-batch, the client computes the training loss and gradient. The current training context, including the gradient norm $n_t = \|g\|_2$, local loss $\ell_t$, and the previous privacy parameters $(\beta_{t-1}, \gamma_{t-1})$, is passed to $\text{ANC}_k$ (Line 9). The ANC outputs updated privacy parameters $(\beta_t, \gamma_t)$ for the current step. The gradient is clipped using the bound $C_{\text{clip}}$ to ensure sensitivity is bounded. Gaussian noise with variance $(\beta_t C_{\text{clip}})^2$ is then injected into the top-$\lceil \gamma_t d \rceil$ entries of the clipped gradient, where $d$ is the model dimensionality. The perturbed gradient is used to update the local model.

After local training, each client constructs a binary mask $M_k$ to mark the gradient coordinates affected by sparse noise injection. To reduce communication overhead, each client transmits only the masked model $W_k \odot M_k$ to the server, effectively implementing a form of communication sparsification. This means that only the top-$k$ most informative gradient components are shared, aligning the communication strategy with the selective perturbation mechanism. The server aggregates the received updates using a sparsity-aware strategy:

$$W_{r+1} \leftarrow \frac{\sum_{k \in S_r} w_k \cdot (W_k \odot M_k)}{\sum_{k \in S_r} w_k \cdot M_k}, \text{ where } w_k \leftarrow \frac{n_k}{\sum_{j \in S_r} n_j}. \tag{13}$$

## 4 Theoretical Analysis

This section presents theoretical guarantees for the proposed FEDANC framework. We begin by stating the key assumptions, which follow standard practices in literature (Lian et al., 2018; Collins et al., 2021; Nguyen et al., 2022; Fu et al., 2022; Xiong et al., 2024; Wang et al., 2024).

**Assumption 1** (Smoothness). Let $\mathcal{L}_k : \mathbb{R}^d \to \mathbb{R}$ be $L$-smooth. That is, for all $\theta, \theta' \in \mathbb{R}^d$, we have:

$$\|\nabla \mathcal{L}_k(\theta) - \nabla \mathcal{L}_k(\theta')\| \leq L\|\theta - \theta'\|.$$

**Assumption 2** (Bounded Noise Variance). Let $g_t$ denote the clipped local gradient and $\widetilde{g}_t$ the corresponding noisy gradient after sparse noise injection. The added noise satisfies:

$$\mathbb{E}\|\widetilde{g}_t - g_t\|^2 \leq (\beta_{\max} C)^2 \gamma_{\max} d,$$

where $C$ is the clipping bound, and $\beta_{\max}, \gamma_{\max}$ are the maximum values.

**Assumption 3** (Bounded Privacy Parameters). The privacy parameters generated by the ANC module are bounded as follows:

$$0 < \beta_{\min} \leq \beta_t \leq \beta_{\max}, \quad 0 < \gamma_t \leq \gamma_{\max}, \quad \forall t.$$

**Theorem 1** (Convergence Guarantee). *Suppose that Assumptions 1–3 hold, and let the learning rate satisfy $\eta = O(1/L)$. Then the following results hold:*

- *Convex case:*

$$\mathbb{E}[\mathcal{L}(\theta^R)] - \mathcal{L}^* = O\left(\frac{1}{R} + \frac{(\beta_{\max} C)^2 \gamma_{\max} d}{R}\right).$$

- *Non-convex case:*

$$\frac{1}{R}\sum_{t=0}^{R-1}\mathbb{E}\|\nabla\mathcal{L}(\theta^t)\|^2 = O\left(\frac{1}{\sqrt{RB}} + \frac{(\beta_{\max}C)^2\gamma_{\max}d}{R}\right),$$

  *where $R$ denote the total number of communication rounds between the server and clients.*

Theorem 1 shows that FEDANC achieves standard sublinear convergence rates under both convex and non-convex objectives. We now turn to the analysis of the corresponding privacy guarantees.

**Theorem 2** (Privacy Guarantee). *At each communication round $t$, Gaussian noise with variance $(\beta_t C)^2$ is added to the top-$k$ coordinates of the clipped gradient, where $k = \gamma_t d$. This guarantees $(\epsilon_t, \delta)$-differential privacy with:*

$$\epsilon_t = \frac{\sqrt{2\gamma_t d \ln(1.25/\delta)}}{\beta_t}.$$

*Using the Moments Accountant (Abadi et al., 2016), the cumulative privacy loss after $R$ rounds is bounded by:*

$$\epsilon_R = O\left(\frac{1}{\beta_{\min}}\sqrt{R\gamma_{\max}d\ln(1.25/\delta)}\right).$$

Theorem 2 confirms that FEDANC ensures formal differential privacy. We next extend the convergence analysis to account for the effect of the privacy-loss regularization term in Equation equation 12.

**Corollary 1** (Convergence with Privacy Regularization). *Let $G_p$ be an upper bound on the gradient norm of the privacy-loss regularization term $L_{\text{privacy}}^{(t)}$. Under the same conditions as in Theorem 1, the following bounds hold:*

- *Convex case:*

$$\mathbb{E}[\mathcal{L}(\theta^R)] - \mathcal{L}^* = O\left(\frac{1}{R} + \frac{(\beta_{\max}C)^2\gamma_{\max}d}{R} + \lambda^2 G_p^2\right).$$

- *Non-convex case:*

$$\frac{1}{R}\sum_{t=0}^{R-1}\mathbb{E}\|\nabla\mathcal{L}(\theta^t)\|^2 = O\left(\frac{1}{\sqrt{RB}} + \frac{(\beta_{\max}C)^2\gamma_{\max}d}{R} + \lambda^2 G_p^2\right).$$

*Moreover, if the regularization weight satisfies $\lambda = O(1/\sqrt{R})$, the impact of the privacy term decays over time, allowing the controller to remain effective without hindering convergence.*

## 5 EXPERIMENTS

### 5.1 EXPERIMENTAL SETUP

**Datasets and models.** We evaluate the effectiveness of our defense mechanisms against gradient inversion attacks using three publicly available datasets: CIFAR-10 (Krizhevsky et al., 2009), FashionMNIST (Xiao et al., 2017), and HARBox (Ouyang et al., 2021). To assess reconstruction quality comprehensively, we also use three datatsets. CIFAR-10 is paired with a randomly initialized ResNet-18 (He et al., 2016), while FashionMNIST is trained with LeNet (LeCun et al., 1998). To evaluate training dynamics under privacy protection, we include additional experiments on FashionMNIST and HARBox. For HARBox, which consists of time-series sensor data, we adopt a hybrid CNN-LSTM architecture (Kim et al., 2016). All datasets are partitioned across $N = 20$ clients using a Dirichlet distribution with the concentration parameter $\alpha$, which controls the degree of statistical heterogeneity. Smaller $\alpha$ values yield more skewed, non-IID data distributions. Following prior work (Wang et al., 2020; Dai et al., 2022; Cao & Gong, 2022; Oh et al., 2022; Chen & Chao, 2022), we set $\alpha = 0.5$ unless otherwise specified.

**Settings and baselines.** Each training round involves 10 randomly selected clients out of 20. For FashionMNIST, local training is performed for one epoch per round using a batch size of 32 and the

Table 1: Defense performance comparison across different attacks and datasets.

| Dataset | Defense Method | DLG | | | IG | | | GI | | |
|---------|----------------|---------|----------|----------|----------|----------|----------|----------|----------|----------|
| | | MSE ↓ | PSNR ↑ | SSIM ↑ | MSE ↓ | PSNR ↑ | SSIM ↑ | MSE ↓ | PSNR ↑ | SSIM ↑ |
| CIFAR-10 | None | 2e-6 | 50.29 | 0.9998 | 0.0568 | 12.58 | 0.1405 | 0.0085 | 48.02 | 0.8732 |
| | Soteria | 0.0519 | 13.14 | 0.1651 | 0.0582 | 12.51 | 0.1399 | 0.0368 | 16.36 | 0.4933 |
| | CENSOR | 0.0673 | 11.96 | 0.0577 | 0.0705 | 11.65 | 0.0418 | 0.0661 | 11.97 | 0.0510 |
| | FEDANC | 0.0769 | 11.37 | 0.0467 | 0.0773 | 11.36 | 0.0318 | 0.0678 | 11.96 | 0.0594 |
| FMNIST | None | 0.0027 | 22.77 | 0.7457 | 0.1814 | 7.47 | 0.0607 | 0.0139 | 18.95 | 0.6941 |
| | Soteria | 1.0538 | 8.36 | 0.0791 | 0.1921 | 7.23 | 0.0568 | 0.1238 | 9.21 | 0.3082 |
| | CENSOR | 1.2680 | 8.22 | 0.0413 | 0.1840 | 7.38 | 0.0304 | 0.1632 | 7.93 | 0.0521 |
| | FEDANC | 1.2737 | 7.52 | 0.0365 | 0.1986 | 7.07 | 0.0564 | 0.1386 | 8.87 | 0.1441 |
| HARBox | None | 0.0291 | 16.86 | 0.5343 | 0.1253 | 9.41 | 0.0598 | 0.0650 | 11.99 | 0.0580 |
| | Soteria | 1.3023 | 6.07 | 0.0402 | 0.1937 | 7.22 | 0.0566 | 0.1267 | 9.08 | 0.0572 |
| | CENSOR | 1.3472 | 5.41 | 0.0074 | 0.2254 | 7.15 | 0.0540 | 0.2194 | 7.20 | 0.0507 |
| | FEDANC | 1.3790 | 5.23 | 0.0062 | 0.2338 | 7.11 | 0.0509 | 0.2691 | 6.93 | 0.0425 |

SGD optimizer with a learning rate of $\eta = 0.01$. For HARBox, we use five local epochs, a batch size of 32, and the Adam optimizer with $\eta = 0.001$. To ensure reproducibility, all experiments are performed with a fixed random seed (1234) and executed on NVIDIA RTX 4090 GPUs in practice.

We evaluate our method under a strong adversarial setting where the batch size is set to one. This configuration maximizes inversion risk and serves as a worst-case scenario. We consider three representative gradient inversion attacks: *DLG (Deep Leakage from Gradients)* (Zhu et al., 2019) reconstructs inputs by minimizing the Euclidean distance between dummy and real gradients, using the L-BFGS optimizer with 300 steps. *IG (Inverting Gradients)* (Geiping et al., 2020) replaces the distance metric with cosine similarity and optimizes dummy inputs using Adam with a learning rate of 0.1. *GI (GradInversion)* (Yin et al., 2021) initializes dummy data from Gaussian noise and iteratively updates it with Adam for 300 steps to align gradients. To provide a comprehensive comparison, we include two state-of-the-art defense methods as baselines. *Soteria* (Sun et al., 2021) reduces privacy leakage by pruning gradients in fully connected layers using a sensitivity mask that filters out high-risk components. *CENSOR* (Zhang et al., 2025) protects private information by projecting gradients into a subspace orthogonal to the inferred attack space through Bayesian sampling.

**Evaluation metrics.** We adopt three standard metrics to quantify reconstruction quality and privacy leakage. *Mean Squared Error (MSE)* measures pixel-wise differences between original and reconstructed images; higher values indicate stronger privacy. *Peak Signal-to-Noise Ratio (PSNR)* quantifies signal clarity relative to noise; lower values suggest greater distortion and better protection. *Structural Similarity Index Measure (SSIM)* evaluates perceptual similarity based on luminance, contrast, and structure; lower scores reflect reduced visual resemblance and improved privacy.

## 5.2 EXPERIMENTAL RESULTS

**Main results.** Table 1 presents the performance of three representative gradient inversion attacks (DLG, IG, GI) under different defense strategies on three datasets. We follow established practice by evaluating privacy leakage during the first communication round, which prior studies (Zhu et al., 2019; Geiping et al., 2020; Yin et al., 2021) have consistently identified as the most privacy-critical phase. This setting serves as a strong proxy for worst-case leakage and is widely adopted to assess the effectiveness of privacy defenses. As expected, all defense methods reduce the effectiveness of attacks compared to the unprotected baseline. Among the baselines, FEDANC consistently achieves strong protection, outperforming Soteria and matching or exceeding CENSOR in most cases. For example, under the DLG attack on CIFAR-10, FEDANC reduces SSIM from 0.9998 to 0.0467, effectively eliminating structural similarity between reconstructed and original inputs. On FashionMNIST and HARBox, where reconstructions are inherently more ambiguous, FEDANC maintains competitive performance, achieving lower MSE and SSIM than CENSOR across several attacks.

These results confirm the effectiveness of FEDANC in defending against gradient inversion across different datasets and model architectures. The adaptive noise controller enables clients to adjust privacy parameters based on real-time training feedback, while the sparse noise injection mechanism selectively perturbs the most informative gradient entries. This combination enhances privacy protection without significantly compromising model performance.

**Convergence evaluation.** As shown in Figure 6, FEDANC converges faster and more stably than static DP baselines, demonstrating its ability to maintain learning efficiency under privacy constraints.

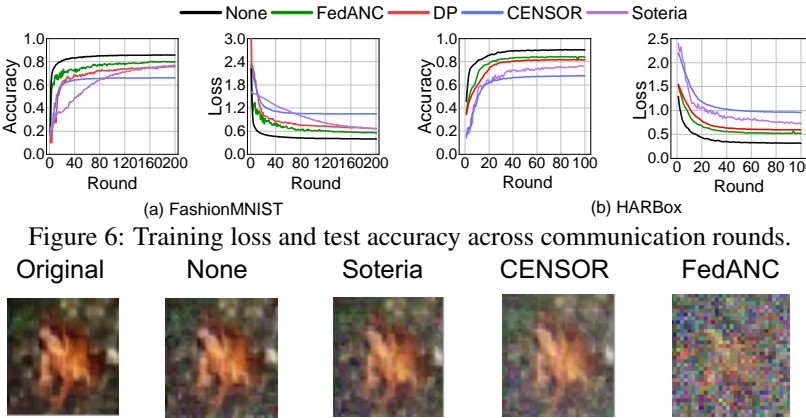

Figure 6: Training loss and test accuracy across communication rounds.

Figure 7: Visual comparison of reconstructed images under the DLG attack.

On FashionMNIST, Soteria removes critical parameters in LeNet through aggressive pruning, while CENSOR disrupts optimization by distorting gradient directions, both leading to slower convergence and reduced accuracy. On HARBox, Soteria weakens output representations, and CENSOR damages LSTM temporal dependencies, causing significant performance degradation. In contrast, FEDANC adaptively adjusts privacy parameters and applies targeted noise, effectively preserving gradient signals and enabling efficient training across different model architectures.

**Qualitative evaluation.** As shown in Figure 7, we qualitatively evaluate the visual reconstruction quality under different defense strategies using the DLG attack on the CIFAR-10 dataset. Without any defense, the original image can be almost perfectly recovered. Soteria and CENSOR introduce moderate distortion, yet the reconstructed images remain partially recognizable. In contrast, our proposed DPSA-FL strategy leads to highly distorted reconstructions with severe noise, making the semantic content unrecognizable.

**Sensitivity to $\lambda$.** We investigate the impact of the privacy-loss regularization strength by varying the coefficient $\lambda$ in Equation 12. As shown in Figure 5(a), model accuracy decreases as $\lambda$ increases. This trend reflects the intended effect of the regularization: a larger $\lambda$ forces the ANC to more strictly align the learned privacy parameters with the target budget $\epsilon_{\text{target}}$. To reduce the deviation penalty, the controller tends to generate more conservative parameters, which increases the degree of gradient perturbation and leads to lower utility. In contrast, a smaller $\lambda$ allows the model to prioritize task performance, potentially at the expense of privacy.

**Impact of $N$.** We investigate the impact of the total number of clients $N$ on global model performance. As shown in Figure 5(b), the model reaches its highest accuracy when $N = 20$, but accuracy gradually degrades as $N$ increases. This decline results from two factors: each client contributes fewer data samples per round, and statistical heterogeneity across clients becomes more pronounced. Together, these effects lead to sparser and more diverse training signals, which slow down convergence under lo-

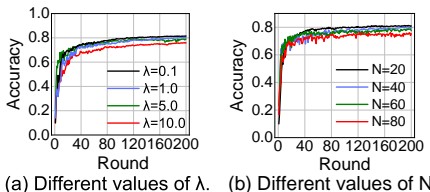

(a) Different values of λ. (b) Different values of N.

Figure 5: Accuracy under different values of $\lambda$ and $N$

cal privacy constraints. Based on these observations, we set $N = 20$ in the main experiments, as this choice offers a balanced level of heterogeneity while ensuring fair comparison across methods.

## 6 CONCLUSIONS AND LIMITATIONS

In conclusion, we presented FEDANC, an adaptive federated learning framework that enables personalized differential privacy through dynamic noise control, sparse gradient perturbation, and global budget alignment. The framework employs the proposed adaptive noise controller to adjust privacy parameters based on local training feedback, achieving an improved privacy-utility trade-off. Some limitations remain, such as the reliance on synthetic ranges for controller pretraining, the added computational cost of top-$k$ perturbation, and the limited coverage of large scale federated environments. Future work will focus on improving online adaptability, reducing overhead, and extending the framework to more diverse and realistic scenarios.

## REPRODUCIBILITY STATEMENT

We have made extensive efforts to ensure the reproducibility of our work. The main paper provides detailed descriptions of the proposed algorithm, model architectures, training setup, hyperparameter configurations, data preprocessing steps, and theoretical results. If the paper is accepted, we will release the code on GitHub. During the rebuttal phase, if reviewers request to check the relevant code, we will upload it to an anonymous GitHub repository to facilitate the review process.

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

# A APPENDIX

## A.1 THE USE OF LARGE LANGUAGE MODELS (LLMS)

Large language models were used only as writing assistants to refine grammar and improve clarity of exposition. All technical ideas, methods, experiments, and theoretical results were fully developed and validated by the authors, who take complete responsibility for the content of this paper.

## A.2 SYSTEM WORKFLOW

Algorithm 1 in this appendix presents the complete training workflow of the proposed FEDANC framework. The algorithm extends the standard FedAvg procedure by incorporating adaptive privacy scheduling via the ANC module, sparse noise injection for efficient gradient perturbation, and a privacy-loss regularization term to ensure alignment with global privacy budgets.

At the beginning of each communication round $r$, the server randomly selects a subset of clients $S_r$ according to a sampling ratio $p$ and sends them the current global model $W_r$ (Lines 3–4). Each selected client initializes its local model as $W_k \leftarrow W_r$, and instantiates the local ANC module $\text{ANC}_k$ if it has not been initialized previously (Line 6). Each client performs $E$ local training epochs. For every mini-batch, the client computes the forward loss and corresponding gradient via backpropagation (Line 9). The current training context, including the gradient norm $n_t = \|g\|_2$, local loss $\ell_t$, and the previous privacy parameters $(\beta_{t-1}, \gamma_{t-1})$, is passed to $\text{ANC}_k$ (Line 10). ANC then returns updated privacy parameters $(\beta_t, \gamma_t)$ for the current step (Line 11). The gradient is clipped using the bound $C_{\text{clip}}$ to ensure bounded sensitivity (Line 12). Sparse noise is then injected into the top-$\gamma_t d$ entries of the clipped gradient (Line 13). The resulting perturbed gradient is used to update the local model (Line 14).

Once local training is complete, each client constructs a binary mask $M_k$ to indicate which gradient coordinates were updated. The masked model $W_k \odot M_k$ is sent back to the server (Line 17). The server then performs sparsity-aware aggregation (Line 19), computing a weighted average over non-zero coordinates based on client-specific update masks and dataset sizes. This ensures consistent and communication-efficient model integration.

## A.3 DETAILS FOR THEORETICAL ANALYSIS

### A.3.1 PROOF OF THEOREM 1

*Proof.* We follow the standard analysis of FedAvg with added noise Li et al. (2020); Bottou et al. (2018), and account for sparse noise injection. By $L$-smoothness (Assumption 1), for any round $t$ and

---

**Algorithm 1** FEDANC Framework

---

**Require:** Client set $K = \{1, ..., N\}$, sampling ratio $p$, communication rounds $R$, local epochs $E$,
batch size $B$, clipping bound $C_{\text{clip}}$, pretrained ANC module
**Ensure:** Final global model $W_R$
 1: Initialize global model $W_0$
 2: **for** $r = 0$ to $R - 1$ **do**
 3:     $K_r \leftarrow \max(p \cdot N, 1)$; randomly sample clients $S_r$ of size $K_r$
 4:     Server broadcasts global model $W_r$ to each $k \in S_r$
 5:     **for all** clients $k \in S_r$ **in parallel do**
 6:         Set local model $W_k \leftarrow W_r$; initialize $\text{ANC}_k$ if not already available
 7:         **for** $e = 1$ to $E$ **do**
 8:             **for** each batch $b \in D_k$ **do**
 9:                 Compute forward loss $\ell_t = \ell(f(W_k; x), y)$ and gradient $g = \nabla \ell_t$
10:                 Input $(n_t = \|g\|_2, \ell_t, \beta_{t-1}, \gamma_{t-1})$ into $\text{ANC}_k$
11:                 Obtain updated privacy parameters $(\beta_t, \gamma_t)$
12:                 Clip gradient: $g' = g \cdot \min(1, C_{\text{clip}}/\|g\|_2)$
13:                 Apply sparse noise injection to obtain $g_{\text{noisy}}$
14:                 Update local model: $W_k \leftarrow W_k - \eta \cdot g_{\text{noisy}}$
15:             **end for**
16:         **end for**
17:         Generate binary mask $M_k$; send $W_k \odot M_k$ to server
18:     **end for**
19:     Server aggregates updates:

$$W_{r+1} \leftarrow \frac{\sum_{k \in S_r} w_k \cdot (W_k \odot M_k)}{\sum_{k \in S_r} w_k \cdot M_k}, \quad w_k = \frac{n_k}{\sum_{j \in S_r} n_j} \tag{14}$$

20: **end for**

---

update $g'_t$,

$$\mathcal{L}(\theta^{t+1}) \leq \mathcal{L}(\theta^t) - \eta \langle \nabla \mathcal{L}(\theta^t), g'_t \rangle + \frac{L\eta^2}{2} \|g'_t\|^2.$$

Taking expectation,

$$\mathbb{E}[\mathcal{L}(\theta^{t+1})] \leq \mathbb{E}[\mathcal{L}(\theta^t)] - \eta \, \mathbb{E}\|\nabla \mathcal{L}(\theta^t)\|^2 + \frac{L\eta^2}{2} \mathbb{E}\|g'_t - \nabla \mathcal{L}(\theta^t)\|^2.$$

We then split $\mathbb{E}\|g'_t - \nabla \mathcal{L}(\theta^t)\|^2$ into

$$\mathbb{E}\|g_t - \nabla \mathcal{L}(\theta^t)\|^2 \; + \; \mathbb{E}\|\widetilde{g}_t - g_t\|^2.$$

The first term is $O(1/B)$ under standard bounded-variance assumptions. The second term is bounded
by Assumption 2:

$$\mathbb{E}\|\widetilde{g}_t - g_t\|^2 \leq (\beta_{\max} C)^2 \, \gamma_{\max} d.$$

Summing for $t = 0, \dots, R - 1$, rearranging, and choosing $\eta = O(1/L)$ gives

$$\frac{1}{R} \sum_{t=0}^{R-1} \mathbb{E}\|\nabla \mathcal{L}(\theta^t)\|^2 = O\Big( \frac{1}{\sqrt{RB}} + \frac{(\beta_{\max} C)^2 \, \gamma_{\max} d}{R} \Big).$$

This yields the non-convex rate. In the convex case,

$$\mathbb{E}[\mathcal{L}(\theta^R)] - \mathcal{L}^* = O\Big( \frac{1}{R} + \frac{(\beta_{\max} C)^2 \, \gamma_{\max} d}{R} \Big).$$

$\square$

### A.3.2 PROOF OF THEOREM 2

*Proof.* At round $t$, we clip the gradient to norm $C$ and select $k = \gamma_t d$ coordinates. We add noise $\xi \sim \mathcal{N}(0, (\beta_t C)^2 I_k)$ to those coordinates. The $L_2$ sensitivity is

$$\Delta_2 = C\sqrt{k} = C\sqrt{\gamma_t d}.$$

By the Gaussian mechanism Dwork & Roth (2014), it guarantees $(\epsilon_t, \delta)$-DP with

$$\epsilon_t = \frac{\Delta_2}{\sigma}\sqrt{2\ln\frac{1.25}{\delta}} = \frac{\sqrt{2\gamma_t d \ln(1.25/\delta)}}{\beta_t}.$$

Then using the Moments Accountant Abadi et al. (2016), the cumulative privacy loss after $R$ rounds is

$$\epsilon_R \leq O\Big(\sqrt{\sum_{t=1}^{R}\epsilon_t^2}\Big) = O\Big(\frac{1}{\beta_{\min}}\sqrt{R\,\gamma_{\max}d\ln\frac{1.25}{\delta}}\Big),$$

where $\beta_t \geq \beta_{\min}$ and $\gamma_t \leq \gamma_{\max}$ for all $t$. $\qquad\square$

