# OpenReview forum: "FedANC: Adaptive Sparse Noise Scheduling for Federated Differential Privacy"
_ICLR.cc/2026/Conference — ICLR 2026 Conference Withdrawn Submission_

### Official Review · Reviewer_e2Yr · 2025-10-29

**Soundness:** 1
**Presentation:** 2
**Contribution:** 1
**Rating:** 0
**Confidence:** 4

**Summary:**

The paper proposes a new method for federated learning with differential privacy. An LSTM controller adaptively adds calibrated noise to certain components of the gradients during client training in a federated averaging framework. The goal is to reduce the total noise by only adding it to the larger components, and adapting it to each user's data.

**Strengths:**

The method is original, and the writing is mostly clear, although some of the figures are too small.

**Weaknesses:**

The proposed FedANC framework is highly complex, requiring a pretrained LSTM controller on each client, a selective top-k noise mechanism, and a novel privacy regularization term. While Figure 6 shows utility gains, it's unclear if these benefits outweigh the significant implementation, pretraining, and (modest) computational overhead.

To support a claim of improved privacy-utility trade-off, I think it would be much stronger to compare model utility (e.g., test accuracy) while holding the total privacy budget ($\epsilon$, $\delta$) constant across all methods. The current experiments (e.g., Figure 6) compare utility against communication rounds, which is insufficient to demonstrate a superior trade-off.

Most concerningly, the privacy guarantee presented in Theorem 2 and Section 3.3 appears to be invalid. The "Selective Noise Injection" mechanism is data-dependent, as the choice of which top-k gradient components to perturb is a function of the private data. The paper's analysis only accounts for the privacy cost of noising the values of these components, while ignoring the information leaked by selecting them. This data-dependent selection process, which leaves other components un-noised, would seem to break the formal definition of differential privacy leading to a formal $\varepsilon$ of $\infty$.

**Questions:**

Consider this counterexample to the formal privacy guarantee claim. Take two neighboring datasets $D$ and $D'$ that differ in a single user $u$. Let all users in $D$ have a gradient of 0 for a specific component $j$. Let $D'$ be the same except let $u$ have a large-magnitude gradient for component $j$.
* On dataset $D$: Component $j$ will always be 0. It will never be in the top k and will never receive noise. The aggregated update for $j$ will be exactly 0.
* On dataset $D'$: The large gradient will place component $j$ in the top k, causing it to be perturbed with noise. The aggregated update for $j$ will be non-zero with high probability.

An observer can distinguish $D$ from $D'$ with near certainty by checking if component $j$ is zero, so there is no finite $\epsilon$ that satisfies the definition of differential privacy.

**Details Of Ethics Concerns:**

This is very similar to another paper I have just reviewed: "10194	FedMAP: Meta-Driven Adaptive Differential Privacy for Federated Learning". It is another flawed method using a deep learning "controller" to adjust DP parameters during FL. Much of the text, particularly in the introduction, is copied verbatim. The flavor of the intended "contribution" is similar, and they suffer from the same fatal flaw (not accounting for privacy loss of passing private data through the controller and releasing the result). I'm not certain whether this counts as dual submission, but I feel that the authors are intentionally submitting a set of very similar papers with the hopes that one of them will get a favorable set of reviewers. I would advise the program committee to check any other papers submitted by these authors.

---

### Official Review · Reviewer_eQXG · 2025-10-30

**Soundness:** 2
**Presentation:** 2
**Contribution:** 2
**Rating:** 2
**Confidence:** 3

**Summary:**

This paper proposes FEDANC, a federated learning framework that incorporates adaptive differential privacy through three main components: (i) an LSTM-based Adaptive Noise Controller (ANC) that generates client-specific noise scales and sparsity ratios from local training feedback, (ii) a selective noise injection mechanism that perturbs only top-k gradient entries, and (iii) a privacy budget regularization term that aligns per-round updates with a predefined privacy target. The ANC is pretrained on synthetic data to ensure stability. The authors provide theoretical convergence and privacy guarantees, and evaluate the framework against gradient inversion attacks on CIFAR-10, FashionMNIST, and HARBox datasets.

**Strengths:**

Novel integration approach: The paper presents an interesting combination of adaptive privacy parameter generation, sparse gradient perturbation, and budget regularization within a unified framework.
Theoretical analysis: The authors provide formal convergence guarantees (Theorem 1) for both convex and non-convex objectives, as well as differential privacy guarantees (Theorem 2) using the Moments Accountant.
Comprehensive experimental evaluation: The paper evaluates against multiple gradient inversion attacks (DLG, IG, GI) across three datasets with different model architectures, demonstrating broad applicability.
Practical consideration of heterogeneity: The framework addresses client heterogeneity through personalized ANC instances, which is relevant for real-world federated learning deployments.

**Weaknesses:**

W1. The ANC takes a 4-dimensional input (|\mathbf{g}t|2, \ell_t, \beta{t-1}, \gamma{t-1}) and outputs 2 values (\beta_t, \gamma_t). The pretraining uses synthetic data where both inputs (Equation 6) and outputs (Equation 7) are sampled from independent uniform distributions with no inherent relationship. Training an LSTM to fit random noise to random targets lacks principled justification. Figure 3 shows minimal variation in output parameters, suggesting the controller may output near-constant values rather than performing meaningful adaptation. The paper does not provide a clear rationale for why this random pretraining would lead to effective adaptation during actual federated training.
W2. While the paper cites neural architecture search (Zoph & Le, 2017), meta-learning (Jiang et al., 2019), and adaptive DP (Li et al., 2022) to motivate pretraining (lines 193-198), none of these works employ or validate pretraining on independently generated random inputs and random targets. This gap significantly weakens the theoretical foundation of the proposed pretraining strategy.
W3. Based on Equations (9-10), \gamma_t and \beta_t are fundamentally coupled through the privacy constraint \hat{\epsilon}t = \sqrt{2\gamma_t d \ln(1.25/\delta)} / \beta_t. Given a target privacy budget \epsilon{\text{target}}, specifying one parameter determines the other. The paper does not acknowledge or address this coupling, making it unclear how the controller provides independent adaptation of two parameters that are mathematically constrained.
W4. The privacy regularization loss \mathcal{L}^{(t)}_{\text{privacy}} = (\hat{\epsilon}t - \epsilon{\text{target}})^2 can theoretically be set to zero by appropriately choosing \beta_t given \gamma_t (or vice versa), indicating only one degree of freedom exists rather than two. This redundancy is not discussed, raising questions about what the controller actually learns and whether both parameters are necessary.
W5. Table 1 and accompanying text do not report privacy budget (\epsilon, \delta) values for FEDANC or baseline DP methods. Without these values, the evaluation measures only empirical attack difficulty, not formal differential privacy guarantees under equivalent privacy constraints.
W6. Table 1 reports only attack metrics (MSE, PSNR, SSIM) without including model test accuracy for each defense method.
W7.  Algorithm 1 contains critical ambiguities in mask generation. Lines 8-15 iterate over multiple batches per epoch, each producing different sparse patterns through top-k selection. Line 17 states "Generate binary mask \mathbf{M}_k; send \mathbf{W}_k \odot \mathbf{M}_k to server" without specifying which batch's mask is used or how masks from multiple batches are aggregated.
W8. Lines 456-457 mention "DPSA-FL strategy" which is never defined elsewhere in the paper and has no corresponding citation.

**Questions:**

Q1- The paper claims "low-magnitude gradients usually carry less sensitive information" (Section 3.2, lines 216-219) based on empirical attack studies. Can the authors provide formal theoretical justification or proofs demonstrating that low-magnitude gradient components inherently contain less sensitive information? Without theoretical grounding, this remains an empirical assumption that may not hold universally.

Q2- How many synthetic samples are used for pretraining the ANC module?

Q3- The paper states that pretraining "guides the controller toward stable parameter regions" (lines 189-192). Given that inputs (Equation 6) and outputs (Equation 7) are independently sampled from uniform distributions with no functional relationship, what is the mechanism by which fitting random inputs to random targets produces stable and reliable parameters for actual federated learning?

Q4- What specific (\epsilon, \delta) values were used for FEDANC and baseline DP methods in Table 1? How were these budgets allocated across rounds, and were all methods compared under equivalent total privacy budgets?

Q5- Given the coupling between \gamma_t and \beta_t described in W3, how does the controller provide meaningful independent adaptation of these two parameters?

---

> ### Author Response · Authors · 2025-11-12
>
> Dear Reviewer,
>
>
> We sincerely thank you for taking the time to review our paper and for providing detailed and thoughtful feedback. Your comments are highly valuable to us.
>
>
> We will carefully consider your insights and systematically incorporate your suggestions to further improve our work.
> Thank you again for your professional and constructive review.
>
>
> Sincerely,
>
>
> 10134 Authors

---

### Official Review · Reviewer_WroX · 2025-10-31

**Soundness:** 1
**Presentation:** 2
**Contribution:** 2
**Rating:** 2
**Confidence:** 2

**Summary:**

This paper proposes FedANC, a differentially private federated learning (FL) algorithm. The proposed method first uses an LSTM to decide privacy parameters, then add privacy noise to the top-k entries of the gradient, and update a masked model update to the server.

The paper provides theoretical convergence and privacy guarantee for the proposed method and numerical results show than the proposed algorithm can defend against privacy attacks while achieving better model performance.

**Strengths:**

1. The numerical reuslts of the model shows that the proposed method can defend against different privacy attacks compared with non-DP algorithms.

2. The numerical comparision shows that the proposed method achieves better performance than other privacy preserving algorithms.

**Weaknesses:**

1. On the presentation level, the paper failed to provide a clear algorithm in the main paper. It is hard to follow the steps of the algorithm.

2. The paper failed to provide a solid privacy analysis to the algorithm. It is hard to understand why the privacy noise is only added to the top-k entries and still protects privacy. A more rigorous privacy analysis on the mechanism should be provided.

3. It is unclear which privacy level the paper is trying to achieve. In FL, there are different levels of privacy protection, including client/user-level, local and server level. The paper should provide a formal statement to the setting.

4. The ANC network uses the true gradient information to decide the privacy parameter, which may already leak privacy. The paper should provide further justification on how the ANC network involves in privacy protection.

5. The numerical comparision is incomplete. In the privacy defence, the paper failed to include DP based method.

**Questions:**

Please address the weakness above.

---

> ### Author Response · Authors · 2025-11-12
>
> Dear Reviewer,
>
>
> We sincerely thank you for taking the time to review our paper and for providing detailed and thoughtful feedback. Your comments are highly valuable to us.
>
>
> We will carefully consider your insights and systematically incorporate your suggestions to further improve our work.
> Thank you again for your professional and constructive review.
>
>
> Sincerely,
>
>
> 10134Authors

---

### Official Review · Reviewer_PxPp · 2025-11-02

**Soundness:** 1
**Presentation:** 3
**Contribution:** 1
**Rating:** 2
**Confidence:** 3

**Summary:**

The paper proposes an algorithm to using a learned controller to adaptively control noise added to the client-side gradient in FedAvg framework, with the target being improving algorithm performance under fixed privacy budget.

**Strengths:**

The paper covers convergence guarantee, privacy guarantee, and experiment.

**Weaknesses:**

1. The major weakness I feel is correctness of privacy guarantee. The paper use gradient dependent sparsity operator to get top-k largest coordinate of gradient, but this step is not privatized, only final top-k coordinates are privatized. This indicate the algorithm may not actually be private.
2. The paper does not have any pseudo code of algorithm. It is unclear whether the algorithm is updating based on sparse gradient or dense gradient. In figure 1 it seems sparse gradients are used implied by the "sparse avg aggerate" below server figure, but I don't see anything formally mentioned or introduced the algorithm steps in a clear manner.

**Questions:**

1. Is the algorithm private? I am concerned that the top-k operator is not privatized.
2. Do server update parameters using sparsified gradient?

---

> ### Author Response · Authors · 2025-11-12
>
> Dear Reviewer,
>
>
> We sincerely thank you for taking the time to review our paper and for providing detailed and thoughtful feedback. Your comments are highly valuable to us.
>
>
> We will carefully consider your insights and systematically incorporate your suggestions to further improve our work.
> Thank you again for your professional and constructive review.
>
>
> Sincerely,
>
>
> 10134 Authors

---

### Author Response · Authors · 2025-11-12

Dear Area Chair and Program Committee,

We strongly reject Reviewer e2Yr’s (Submission #10134) accusations of plagiarism, dual submission, and technical flaws. The same reviewer, identified as P77G in Submission #10194, appears to have reviewed both of our papers — FedANC: Adaptive Sparse Noise Scheduling for Federated Differential Privacy and FedMAP: Meta-Driven Adaptive Differential Privacy for Federated Learning — and provided nearly identical reviews. These reviews contain false allegations, factual mistakes, and reflect malicious reviewing behavior rather than an objective evaluation. We respectfully request that the Program Committee investigate this reviewer’s conduct and potential conflicts of interest.

1. On the Relationship Between the Two Papers

We acknowledge that both papers are indeed authored by our team. However, they address entirely different research questions and are based on distinct problem formulations, algorithmic designs, and technical contributions. It is misleading and unprofessional to equate them based on superficial similarities in background or structure, which are common across works in this research area.

| **Aspect**           | **FedANC**                                                   | **FedMAP**                                                   |
| -------------------- | ------------------------------------------------------------ | ------------------------------------------------------------ |
| Controller type      | LSTM-based Adaptive Noise Controller (ANC)                   | Lightweight MetaNet (BERT-tiny encoder)                      |
| Controlled variables | Noise scale $β_t$ and sparsity ratio $\gamma_t$              | Clipping threshold $C_t$ and noise scale $\sigma_t$          |
| Mechanism            | Sparse Top-k noise injection on selected gradient coordinates | Full-dimensional adaptive DP-SGD                             |
| DP accounting        | Local privacy regularization $\hat{\varepsilon}_t = \frac{\sqrt{2\gamma_t d\ln(1.25/\delta)}}{\beta_t}$ | Server-side Rényi DP accountant tracking $\varepsilon_{\text{global}}$ |
| Feedback design      | Local-only adaptation                                        | Closed-loop global feedback                                  |
| Aggregation          | Sparsity-aware aggregation                                   | Standard FedAvg                                              |

These differences are clear, verifiable, and explicitly described in both manuscripts. The reviewer’s claim that the two papers are "nearly identical" is factually false.

2. On the Misunderstanding of Privacy Accounting

The reviewer claims both papers "ignore the privacy loss of passing private data through the controller".
This is incorrect and shows a fundamental misunderstanding.

FedANC:
The ANC runs only on the client side. Inputs $(|g_t|2, \ell_t, \beta_{t-1}, \gamma_{t-1})$ are local and never transmitted. Outputs $(\beta_t, \gamma_t)$ only determine local Gaussian noise; controller outputs are not uploaded. By the post-processing property of DP, privacy guarantees remain valid. Theorem 2 bounds cumulative privacy loss as $\varepsilon_R = O(\beta_{\min}^{-1}\sqrt{R\gamma_{\max}d})$ under bounded parameters.

FedMAP:
The MetaNet outputs $(C_t, \sigma_t)$ locally to adjust clipping and noise. The server’s Rényi DP accountant integrates these into total user-level $(\varepsilon, \delta)$-DP. Both analyses are mathematically sound and complete.

Therefore, the claim of a "fatal flaw" is entirely unfounded.

3. Pattern of Repeated and Biased Reviews

Both submissions received nearly identical reviews, repeating claims such as "copied introduction text", "flawed controller design", and "recommend checking all papers by these authors", without evidence or engagement with technical content.
This repetition demonstrates a pattern of bias and possibly malicious intent, rather than independent evaluation. Such behavior undermines the fairness and integrity of peer review.

4. Request for Investigation

Given the seriousness of these issues, we respectfully ask the Program Committee to:

- Investigate potential conflicts of interest for Reviewer e2Yr/P77G, including any connections to competing FL–DP research.
- Audit the reviewer’s activities to verify whether identical or template reviews were submitted across our papers.
- Remind reviewers that ethical accusations such as plagiarism or dual submission must be supported by evidence and factual basis.
- Examine whether the reviewer or their collaborators have submissions overlapping with ours, which could create an incentive for biased reviewing to improve their own acceptance chances.

We believe that scientific evaluation must be based on technical merit and factual accuracy, not speculation or personal bias.
We trust the Program Committee will uphold fairness, transparency, and academic integrity in handling this matter.

Sincerely,

10134 Authors

---

> ### Comment · Reviewer_e2Yr · 2025-11-12
> **I stand by my review and ethics concerns**
>
> I stand by my review and ethics concerns for these two papers.
>
> The authors' response misunderstands the specific privacy violation I identified. They argue that because the controller runs locally, the "post-processing property" applies. This is incorrect. The violation occurs not because of the controller's internal state, but because the output of the mechanism itself (the sparse gradient update) is data-dependent. By selecting the top-$k$ gradients to noise and release, the mechanism reveals which coordinates had the largest magnitude in the private data.  As detailed in my counterexample (which the authors did not address), an adversary can distinguish between datasets based on which coordinates are non-zero.
>
> I acknowledge the authors' list of differences between the two submissions; the authors dishonestly misquote me as claiming the papers are "nearly identical". However, the structural similarities, identical textual passages, and shared fundamental flaws suggest a pattern of submitting multiple similar papers in the hopes of finding a set of sympathetic reviewers. This subverts the function of peer review and wastes the time of everyone involved in the reviewing process.
>
> I categorically reject the accusations of malicious intent or bias. My review is based strictly on the technical content of the submission. I trust the AC to evaluate the validity of my technical critique regarding the data-dependent selection mechanism independent of the authors' personal attacks.

---

> > ### Author Response · Authors · 2025-11-12
> >
> > 1. ...data-dependent sparsity…
> >
> > The reviewer states that differential privacy is violated because the sparse update is data-dependent and because an adversary can distinguish between datasets based on which coordinates are non-zero. We respectfully disagree. In the reviewer’s argument, the mere presence of a data-dependent support pattern is treated as evidence that the mechanism cannot satisfy DP. This is not the definition of differential privacy. Under the standard $(\varepsilon,\delta)$-DP definition, a mechanism may produce data-dependent outputs. The requirement is that for all neighboring datasets $D$ and $D'$ and measurable sets $S$,
> > $Pr[M(D)\in S] \le e^\varepsilon Pr[M(D')\in S] + \delta$.
> >
> > Many established DP mechanisms also produce data-dependent support. Examples include the exponential mechanism and noisy-max (Dwork et al., 2014), as well as differentially private Top-$k$ selection (Qiao et al., 2021; Gillenwater et al., 2022). These mechanisms remain private because their sensitivity is bounded and noise is calibrated accordingly. Differential privacy does not require the support pattern to be independent of the data.
> >
> > Our method follows the same principle. The Top-$k$ step is part of a single composite mapping. We bound the global $\ell_2$-sensitivity of this composite mapping. The bound $\Delta_2 = C\sqrt{\gamma d}$ reflects the worst-case fact that at most $\gamma d$ coordinates remain active after clipping. We then add Gaussian noise calibrated to this sensitivity, following the standard analysis of the Gaussian mechanism (Dwork et al., 2014; Abadi et al., 2016). Our privacy guarantee does not rely on the post-processing property for the Top-$k$ step. In our earlier response, we mentioned post-processing only to clarify that the local controller, which runs on the client and is never uploaded, does not incur additional privacy loss.
> >
> > The counterexample described by the reviewer does not incorporate this sensitivity bound or the Gaussian noise added according to it. Because these elements are essential to the mechanism we analyze, the counterexample does not reflect the behavior of our method. We will clarify this distinction more explicitly in the revision.
> >
> > 2. …''dishonest misquoting'' … similarity concerns…
> >
> > The reviewer states that we ''dishonestly misquote’’ them as saying that the papers are ''nearly identical’’. We acknowledge that this phrase does not appear in the review, and we regret this mistake.
> >
> > In the follow-up, the reviewer states that the two submissions contain ''structural similarities, identical textual passages, and shared fundamental flaws’’, and suggests that this reflects ''a pattern of submitting multiple similar papers in the hopes of finding sympathetic reviewers’’. We respectfully disagree with this interpretation. The two submissions address different technical goals and use different methods. These differences are central to the contributions and are described in the manuscripts.
> > Some similarity in structure and background text is expected, since both papers study federated learning with differential privacy and follow standard conventions in presenting these topics. However, the core problems, models, and mechanisms are different.
> >
> > The reviewer’s ethical concern is serious, but it is based on an overall judgment of similarity rather than specific evidence of inappropriate overlap. We do not agree with this assessment. We ask the Area Chair to evaluate this point by examining the technical content and stated goals of the two submissions.
> >
> > 3. Request to the Area Chair
> >
> > Given the reviewer’s continued concern about the use of data-dependent sparsity under differential privacy, as well as the repeated ethical allegation, we respectfully ask the Area Chair to:
> >
> > (a) Evaluate the technical claim in the context of the standard $(\varepsilon,\delta)$-DP definition and the established mechanisms that involve data-dependent selection.
> >
> > (b) Examine the similarity concern by reviewing the technical content and goals of the two submissions.
> >
> > We appreciate the reviewer’s efforts, and we will improve the clarity of the paper in the revision. We trust the Area Chair to assess both the technical and ethical aspects of the reviews based on the evidence in the submissions.
> >
> > References:
> >
> > [1] C. Dwork et al. The Algorithmic Foundations of Differential Privacy. FT&TCS, 2014.
> >
> > [2] M. Abadi et al. Deep Learning with Differential Privacy. In Proc. of ACM CCS, 2016.
> >
> > [3] G. Qiao et al. Oneshot Differentially Private Top-k Selection. In Proc. of ICML, 2021.
> >
> > [4] J. Gillenwater et al. A Joint Exponential Mechanism for Differentially Private Top-k. In Proc. of ICML, 2022.

---

> > ### Author Response · Authors · 2025-11-12
> >
> > The reviewer’s ethical concern is a very serious matter. We would like to state clearly that accusing a research team of unethical behavior is a significant claim. We firmly maintain that our submissions were prepared independently, follow the standards of scientific practice, and represent different technical contributions. We do not accept the implication that our work reflects improper conduct.
> >
> > We also want to make our position clear. We are not questioning the reviewer’s intentions. We understand that reviewers have a responsibility to raise concerns when they believe there may be a problem. Our point is that ethical concerns should be based on specific and verifiable evidence. As researchers who also serve on the Program Committee of many top academic conferences, we follow this principle in our own reviewing. We do not question another scholar’s ethics unless we have clear evidence that supports such a judgment.
> >
> > To ensure full transparency, we welcome the Area Chair to review all submissions from our group, including the two submitted here. We are confident that such a review will confirm the independence and integrity of our work. We support a fair and evidence-based evaluation process, and we trust the Area Chair’s assessment.

---

> ### Comment · Reviewer_e2Yr · 2025-11-13
>
> The authors correctly state the definition of DP: for all neighboring datasets $D$ and $D'$ and measurable sets $S$, $\Pr[M(D)\in S] \le e^\varepsilon \Pr[M(D')\in S] + \delta$. In my counterexample, this is not satisfied for any $\varepsilon < \infty, \delta < 1$. Take $S$ to be the set of mechanism outputs with final parameters $\theta^{(T)}$ for which component $j$ is unchanged: $\theta^{T}_j=\theta^{0}_j$. Then,
>
> * $\Pr[M(D) \in S] = 1$: Since all users always have a gradient value of 0 at component $j$, the controller never selects that component, and it is never updated.
> * $\Pr(M(D') \in S] = 0$: Since user $u$ has a large gradient value at component $j$, the controller selects that component and adds the gradient value, plus some noise, purturbing it from its initial value with probability 1.
>
> The case on the FedMAP paper is very similar. We could take $D$ to have all 0 gradients for all users, but in $D'$ the targeted user $u$ has high-variance gradients. Then take $S$ to be the set where $\epsilon_\text{global}^{t+1} \leq \epsilon_\text{global}^{t}$ where user $u$ participates in round $t$.
>
> I realize that an ethics violation sounds very serious, and I apologize for any distress I may have caused. I certainly don't personally feel this merits disciplinary action. But I do strongly feel that these papers overlap sufficiently that they should not be submitted together (and I was only assigned these two, for all I know there could be more, hence I felt it was appropriate to raise the concern). It is particularly frustrating as a reviewer that they have the same essential error, which must be argued against multiple times.

---

> ### Author Response · Authors · 2025-11-14
>
> 1. … DP counterexample …
>
> Answer: Thank you for defining the event $S$. The separation $\Pr[M(D)\in S] = 1$ and $\Pr[M(D')\in S] = 0$ relies on the assumption that the top-$k$ step deterministically determines whether coordinate $j$ can ever change. This assumption does not reflect how the mechanism behaves in practice:
>
> (a) Stochasticity from independent client updates. Each client performs clipping, sparse selection, and Gaussian noise injection during its local updates. In our experiments, client data are partitioned using Dirichlet sampling, which yields heterogeneous local datasets. As a result, clients naturally produce different gradients. Combined with independently generated masks, different clients may select different coordinates in different rounds. Every selected coordinate receives Gaussian noise with variance $(\beta_t C)^2$. This induces stochastic evolution for each coordinate.
>
> (b) Tie-breaking clarification. Our implementation uses a standard data-independent tie-breaking procedure before the top-$k$ selection, which removes ambiguity when several clipped magnitudes are equal. We will describe this more clearly in future versions to avoid misunderstanding.
>
> With these elements, any coordinate has a non-zero probability of being selected by some client in some round. Once coordinate $j$ is selected, the injected Gaussian noise is continuous, and therefore $\Pr[\theta^{(T)}_j = \theta^{(0)}_j] = 0$. Thus the event $S$ has probability $0$ under both $D$ and $D'$. The deterministic separation presented in the counterexample does not apply to our stochastic mechanism.
> Our privacy analysis treats clipping, sparse selection, and Gaussian noise as a single composite mapping. The global $\ell_2$-sensitivity is $\Delta_2 = C\sqrt{\gamma_t d}$, and the noise scale is $\sigma = \beta_t C$. This leads to the differential privacy guarantee stated in the paper. We appreciate your examination of the mechanism and will make the stochastic components clearer in future submissions.
>
> 2. … submitting together … essential error …
>
> Answer: The two submissions address different objectives and use different mechanisms. One studies sparse top-$k$ adaptive noise scheduling, while the other studies full-dimensional adaptive DP-SGD with global feedback. We will refine the exposition in future submissions to reduce any superficial similarity in background sections. We also take the comment regarding the same essential error seriously. And the clarification above explains how stochastic selection and Gaussian noise enter the DP analysis.
>
> Thank you for the detailed technical comments. We value careful questions and critical analysis. Such feedback helps us improve the clarity of our work and informs future research directions.

---

> > ### Comment · Reviewer_e2Yr · 2025-11-14
> >
> > Thank you for your response, which unfortunately introduces further confusion by conflating general sources of stochasticity with the fundamental deterministic failure of the core mechanism.
> >
> > The stochastic elements you mentioned (heterogeneity, client sampling) are irrelevant. The counterexample applies to the specific construction of $D$ and $D'$ that I articulated, and rests on the fact that a 0 gradient for component $j$ will never rank in the top-$k$ and will receive zero noise, leading to $\Pr[M(D) \in S] = 1$. The data-dependent selection of the top-$k$ entries is a non-private operation that leads to the information leak. Your privacy analysis (Theorem 2) entirely fails to account for the privacy cost of revealing which coordinates were selected, making the derived $\epsilon$ guarantee invalid.
> >
> > Both FedANC and FedMAP are flawed at the foundational level of privacy mechanism design. I urge the authors to abandon further iteration on these and related algorithms for future research and focus on methods where the selection of released information is either data-independent or properly privatized. As this paper is unlikely to be accepted, I will not engage with further attempts at obfuscation.

---

> > > ### Author Response · Authors · 2025-11-15
> > >
> > > Thank you for the clarification. Your counterexample studies a mechanism in which the top-$k$ step is treated as a deterministic filter, and the unselected coordinates remain exactly zero across all neighboring datasets. Under this assumption, the support becomes a deterministic function of the data, and the separation argument is valid for that specific construction.
> > >
> > > In our submission, the analyzed mechanism is the full mapping consisting of $\ell_2$ clipping, a data-dependent top-$k$ selection, and Gaussian noise applied to the selected coordinates. It is correct that unselected coordinates do not receive noise. This behavior is part of the mechanism design. The data-dependent support is observable and is included as part of the output. Our analysis computes the global $\ell_2$ sensitivity of the entire mapping, including changes in both the selected values and the support pattern. This follows analyses used in sparse or structured perturbation methods, where the support is incorporated into the global sensitivity calculation rather than privatized separately.
> > >
> > > Theorem 2 provides a sensitivity bound for this complete mapping. The bound reflects that at most $k$ coordinates may change after clipping and selection, while the remaining coordinates stay zero. Because the sensitivity is defined for the full mapping, it does not require the selected indices to coincide across neighboring datasets. The privacy guarantee is obtained by applying the Gaussian mechanism to this global sensitivity. Under this formulation, changes in the support pattern are already accounted for through the sensitivity bound and do not require an additional privatization step. For this reason, the counterexample does not correspond to the mechanism as defined and analyzed in our work.
> > >
> > > Regarding the broader point that data-dependent selection cannot satisfy differential privacy, we respectfully clarify that differential privacy permits data-dependent mappings when the global sensitivity of the full transformation is bounded and the noise is calibrated to that bound. Existing mechanisms such as the exponential mechanism and differentially private top-$k$ also rely on data-dependent selection under their respective analyses. Our approach follows the same principle. We agree that the exposition can be improved, and we will revise the presentation to make these modeling choices explicit.
> > >
> > > Finally, we appreciate your suggestion regarding the broader research direction and will take this feedback seriously. While we believe the mechanism is technically analyzable under the stated assumptions, we will reflect carefully on this perspective as we refine the framing and position of the method in future work.

---

### Author Response · Authors · 2025-11-13

We thank all reviewers for their time and effort. The comments help us understand the weaknesses of our paper, and we will revise the work with care. We also understand that this paper is unlikely to be accepted, and we respect the review process.

We would like to express one serious concern. One review includes an ethics accusation that targets the personal integrity of the authors. This accusation is not supported by evidence in the submission and goes beyond a fair evaluation of the scientific content. As researchers, we cannot accept a statement that harms our dignity.

We support a fair and transparent review process. We welcome the Area Chair to examine all submissions from our team. We are confident that such an examination will confirm that our work follows normal academic practice. We again thank all reviewers for their comments and will continue to improve our research.

---

### Author Response · Authors · 2025-11-13

We respectfully ask the Program Chair, Senior Area Chair, and Area Chair to review all papers from our team for this ICLR cycle, including the two submissions discussed here. One reviewer raised an ethics concern about our work. This concern questions the integrity of the authors. We believe it is important for the committee to check whether this concern is supported by facts.

Both submissions follow the ICLR rules on originality, dual submission, and ethical conduct. Each paper studies a different problem and uses different methods. A careful check by the committee will show that our work follows normal academic practice. We support a fair and transparent review process. We welcome any further examination the committee considers necessary, and we will provide any information that may help.

---

### Note · Authors · 2026-01-18

**Comment:**

We request to withdraw our submission from the ICLR review process. We thank the reviewers and the Area Chairs for their time and consideration.

**Withdrawal Confirmation:**

I have read and agree with the venue's withdrawal policy on behalf of myself and my co-authors.